



# Usability of aerial video footage for 3D-scene reconstruction and structural damage assessment

Johnny Cusicanqui, Norman Kerle, and Francesco Nex

Faculty of Geo-Information Science and Earth Observation (ITC), University of Twente, Enschede, the Netherlands

*Correspondence to:* Johnny Cusicanqui (jfcusicanqui@gmail.com)

**Abstract.**

Remote sensing has evolved into the most efficient approach to assess post-disaster structural damage, in extensively affected areas through the use of space-borne data. For smaller, and in particular, complex urban disaster scenes, multi-perspective aerial imagery obtained with Unmanned Aerial Vehicles and derived dense colour 3D-models are increasingly being used. These type of data allow the direct and automated recognition of damage-related features, supporting an effective post-disaster structural damage assessment. However, the rapid collection and sharing of multi-perspective aerial imagery is still limited due to tight or lacking regulations and legal frameworks. A potential alternative is aerial video footage, typically acquired and shared by civil protection institutions or news media, and which tend to be the first type of airborne data available. Nevertheless, inherent artifacts and the lack of suitable processing means, have long limited its potential use in structural damage assessment and other post-disaster activities. In this research the usability of modern aerial video data was evaluated based on a comparative quality and application analysis of video data and multi-perspective imagery (photos), and their derivative 3D point clouds created using current photogrammetric techniques. Additionally, the effects of external factors, such as topography and the presence of smoke and moving objects were determined by analyzing two different earthquake-affected sites: Tainan (Taiwan) and Pescara del Tronto (Italy). Results demonstrated similar usabilities for video and photos. This is shown by the short 2 cm of difference between the accuracies of video and photo-based 3D Point clouds. Despite the low video resolution, the usability of this data was compensated by a small ground sampling distance. Instead of video characteristics, low quality and application resulted from non-data related factors, such as changes in the scene, lack of texture or moving objects. We conclude that current video data are not only more rapidly available than photos, but they also have a comparable ability to assist in image-based structural damage assessment and other post-disaster activities.



# 1   Introduction

The effectiveness of post-disaster activities during the response and recovery phases relies on accurate and early damage estimations. Post-disaster Structural Damage Assessment (SDA) is typically based on ground surveying methods, the most accurate approach for the assessment and classification of structural building damage. Main response activities, however, rely

on rapid building damage information which is not possible through such ground-based methods. Remote sensing techniques are an important tool for rapid and reliable SDA. Different approaches have been extensively studied; a structured review of all these approaches was presented by Dong and Shan (2013). Traditionally, data obtained from nadir perspective (i.e. vertical) platforms, such as satellites and airplanes were used. However, the lack of oblique perspective of this kind of data hinders the identification of damage-related features at building façades, which are critical for a comprehensive damage assessment

(Gerke and Kerle, 2011; Vetrivel et al., 2015). The potential for SDA of oblique perspective data, such as Pictometry and aerial stereoscopic imagery, has long been realized; however, means to process those data have traditionally been limited to manual/visual inspection, or to single-perspective texture-based damage mapping (Mitomi et al., 2002; Ogawa and Yamazaki, 2000; Ozisik and Kerle, 2004; Saito et al., 2010). A very promising source of oblique, and even multi-perspective post-disaster aerial data that has matured in recent years is Unmanned Aerial Vehicles (UAVs). In comparison to traditional platforms, UAVs

are more versatile in capturing multi-perspective high resolution imagery, are easy to transport and fly, allow easier access to destroyed areas than ground surveys, and provide an economically more attractive solution than traditional airborne approaches (Nex and Remondino, 2014). Additionally, modern photogrammetry and computer vision techniques for image calibration, orientation and matching, allow rapid processing of multi-perspective data for the generation of dense color 3D point clouds (3DPCs) (Vetrivel et al., 2017). State-of-the-art methods based on semantic reasoning and deep learning, have been developed

for highly accurate and automated post-disaster SDAs through the use of multi-perspective imagery and 3DPCs. However, the acquisition of multi-perspective imagery using UAV still cameras, labeled in this research as photos, requires a structured plan during crisis situations, one integrated with all humanitarian assistance regulations, and which is often hindered by tight or lacking regulations and legal frameworks. To date, this poses the most severe limitation for the effective collection and sharing of this kind of data. Alternatively, aerial video footage typically obtained by civil defense forces, police, fire services or news

media is less subjected to the legal regulations. Those data optimize the endurance flight time capacity and can even be streamed straight to the operator to be available in near real-time, making them a potentially powerful information source for damage mapping (FSD, 2016; Xu et al., 2016). Video data are in essence a combination of still frames that can be easily extracted and used as aerial images for post-disaster SDA. However they differ from the ones obtained from either compact or digital single-lens reflex (SLR) cameras due to inherent artifacts (e.g. motion-blur effects) and generally lower quality characteristics

(e.g. resolution and frame redundancy) (Alsadik et al., 2015; Kerle and Stekelenburg, 2004). The effects of such video artifacts and lower quality characteristics, and the true potential of this kind of data for SDA and other post-disaster activities remain poorly understood. This research aims at determining the usability of aerial video data, extracted as frames, for an effective SDA, in comparison to data obtained by UAV-based still cameras, here referred to as photographs or photos. Two study cases





were selected: the 2016 Taiwan earthquake that caused severe but local damage in the city of Tainan, and the 2016 Pescara del Tronto Earthquake in Italy, that instead caused more extensive and diverse (low to complete) damage.

## 1.1 Image and video-based structural damage assessment

Dong and Shan (2013) divided image-based SDA approaches into mono and multi-temporal. The largest group of existing methods corresponds to multi-temporal analysis of data, where change detection is applied to a variety of pre and post-disaster RS data. However, while for satellite data the pre-event archive has been growing, frequently allowing change detection, for airborne data, including very detailed imagery obtained from UAV's, such reference data are usually lacking. Therefore, approaches linked to multi-perspective data are mainly of mono-temporal nature. 2D (i.e. only using multi-perspective imagery) and 3D (i.e. also with UAV-based 3DPCs) UAV data have been used for mono-temporal image-based SDA. 2D image features have been studied for the recognition of damage patterns, among which texture features such as Histogram of oriented Gradients (HoG) and Gabor were found to be the most effective (Samadzadegan and Rastiveisi, 2008; Tu et al., 2016; Vetrivel et al., 2015). Vetrivel et al. (2015) extracted Gabor wavelets and Histogram of Gradient orientation (HoG) texture descriptor features from UAV imagery for a supervised learning classification of damage-related structural gaps, by considering the distinctive damage textural pattern at these types of gaps surroundings. This approach led to high accuracies, but the generalization was limited due to the sensitivity of Gabor wavelets and HoG to scale changes and clutter. An improvement on generalization was achieved later by Vetrivel et al. (2016) who used the same texture features in a more robust framework for feature representation, Visual Bag of Words (BoW). Pattern recognition methods based on texture features are highly capable of discriminating damaged regions; however, they rely on site-specific damage patterns that still limit an appropriate transferability. Domain-specific semantic analysis is another 2D approach to classify structural damage. This approach refers to the use of contextual association of all retrievable features (e.g. textural, spectral, geometric, etc.) for the development of ontological classification schemes (e.g. set of rules), defined based on domain knowledge. Fernandez Galarreta et al. (2015) developed a methodology for the extraction and classification of building segments based on semantic reasoning using UAV imagery. However, the complex and subjective aggregation of segment damage-related information to building level was a main drawback to this approach. Recently, deep learning methods such as Convolutional Neural Networks (CNN) led to a major improvement in SDA accuracies. CNN are networks designed to work with image data and are composed of different groups of layers. Convolutional layers represent the first group, and are a set of filter banks composed of image and contextual feature filters. The following group corresponds to data shrinking and normalization layers. Finally, the last group transforms all the information generated and outputs features with high-level reasoning; usually this layer is connected to a loss function such as Support Vector Machine for classification. Vetrivel et al. (2017) tested different CNN architectures for different sites and data, and obtained highly accurate damage estimations using a pre-trained model tuned with damage and no-damage training samples from a range of disaster sites. Besides, 3D data such as UAV-generated 3DPCs were used for damage estimations, based on the extraction of geometric features. Only a few mono-temporal SDA methods using 3D data have been developed, typically based on point neighborhoods and segment-level extracted geometric features. Khoshelham et al. (2013) tested segment-based features such as point spacing, planarity and sphericity for an accurate classification of damage building roofs. A more semantic approach was developed by



Weinmann et al. (2015), who extracted different geometric features from every 3D point neighborhood and used them for the posterior classification of all 3D points. These methods, however, did not represented well the irregular pattern of damaged areas, hence Vetrivel et al. (2017) instead used a segment-level histogram approach based on HoG, for a better representation of these patterns in every 3D point neighborhood.

Video data have been tested for image-based 3D scene reconstruction in different applications. For example, Singh et al. (2014) used video sequences for 3D City modeling and later Alsadik et al. (2015) and Xu et al. (2016) treated video data for the detection of cultural heritage objects. In general, processing pipelines were focused on an effective selection of good quality and non-redundant video frames and their automated and accurate geo-registration (Ahmed et al., 2010; Clift and Clark, 2016; Eugster and Nebiker, 2007). These advances allow fast and reliable processing of video data in near-real time; however,

less research has been done on their potential on post-disaster SDA. Mitomi et al. (2002) analyzed distinct spectral damage features in individual video frames, obtaining a reasonable building damage classifications using a Maximum likelihood classifier whose efficiency, however, was limited by the need for a continuous training sample selections. Later Ozisik and Kerle (2004), demonstrated that video data are able to improve structural damage estimations once integrated with moderate resolution satellite imagery. However, such data require substantial work to be processed, registered and integrated. A similar, but

more elaborated approach was presented by Kerle and Stekelenburg (2004) who created a 3D environment from oriented and improved video frames. In this research, the automated classification results were poor, highlighting the quality-related limitations of video data common at that time. Video-based SDA approaches show that aerial video data require substantial work to be incorporated into conventional damage classification approaches. However, they can be a potential source of damage-related information due to their multi-perspective nature, similar to UAV images, and real-time availability. Additionally, modern cam-

eras capture video data with an improved resolution (up to 4K), and software resources allow easier video data processing for 3D model generation.

## 2   Methodology

### 2.1   Study cases and data description

Two earthquake-affected urban areas were selected for this research due to their contrasting characteristics, Tainan (Taiwan)

and Pescara del Tronto (Italy; Figure1). Both cities were affected by earthquakes of similar magnitude; 6.4 and 6.2 in February and August 2016 respectively. Tainan is a large city with regular topography where most buildings are composed of reinforced-concrete. Conversely, Pescara de Tronto is a rural city settled on one hillside of the Tronto Valley, hence topography there is hilly and houses are mainly made of stone. These disparate characteristics give each case study different damage features: while in Tainan damage was largely confined to a single collapsed high-rise building, in Pescara del Tronto it was extensive and

highly variable (i.e., also intermediate damage levels can be found). The aim of selecting these contrasting cases is to obtain a broader understanding of video data usability in SDA.

After these earthquake events UAV photos and videos were acquired. UAV-based oblique and nadir photos of six dates after the disaster were collected in Tainan for most of the damaged zone, while a more limited amount of nadir photos was obtained




in Pescara del Tronto (Table 1). Photo resolution in both cases is 12 MP. The Pescara del Tronto photos exhibit some blur effects due to a low shutter speed (i.e. 1/60sec). Nadir photos in both study areas were obtained from larger flying heights, hence Ground Sampling Distance (GSD) is also larger for this data type than for the oblique photos. Likewise, several aerial video stream were obtained at each study site and uploaded to video sharing websites such as Youtube, from where the most

5  complete sequences of highest quality were downloaded. The resolution for both areas was FullHD (i.e. 1920x1080pixels ≈ 2.1 MP). The video streams downloaded present different perspectives and characteristics. Due to the flat topography, the Tainan footage presents a more complete overview of the scene than Pescara del Tronto. However, the footage is frequently affected by extensive smoke emanating from the collapsed building, and was obtained from a higher elevation (also resulting in higher GSD) than the one of Pescara del Tronto.

10  Additionally, for each study area precise 3D Ground Control Points (GCPs) were obtained. In both cases the GCPs are well distributed over the terrain; however, in Tainan GCPs were only measured using street markings, resulting in 3D position data with very limited z-variation.

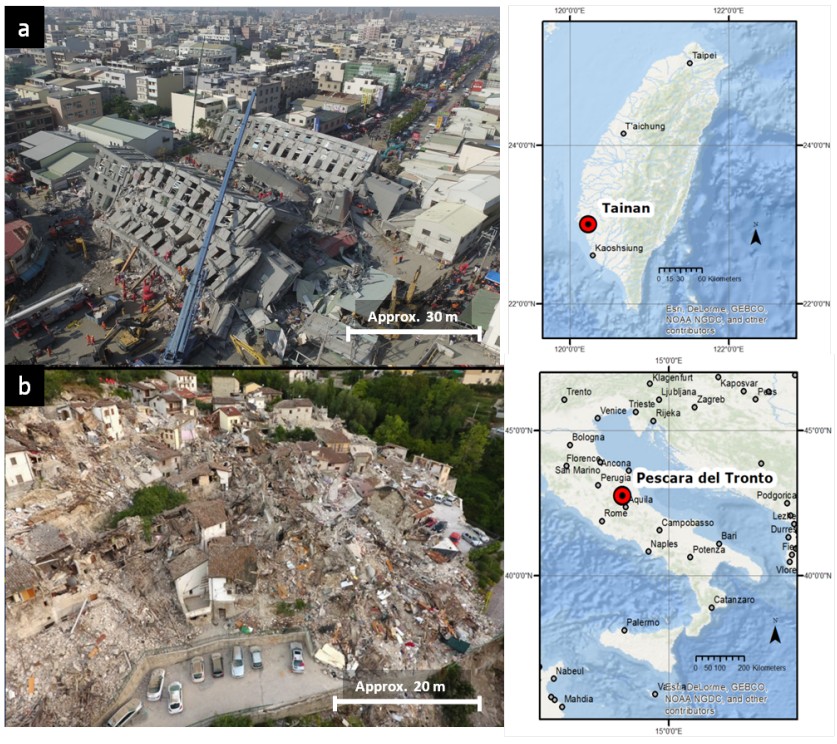

**Figure 1.** Study cases. (a) Taiwan earthquake February, 2016: Flat terrain, smoke presence and highly localized damage (Weiguan Jinglong collapsed building in the middle) (b) Pescara del Tronto earthquake August, 2016: Hilly terrain and more diverse, widespread damage.



| Study Case | Data | Perspective | Image propieties | Approximate altitude (m) | GSD (cm) | Acquision date |
|---|---|---|---|---|---|---|
| Tainan | Photos | Nadir | 4000x3000p (12 MP) | 87 | 11.64 | 7, 10, 11, 12, 13 and 14-02-16 |
| | | Oblique | | 42 | **3.56** | |
| | Video | Oblique | 1920x1080p (2.1 MP) | 50 | 7.66 | 6-02-16 |
| | GCPs | 25 GCPs acquired from street markings in Tainan (limited z-variation) | | | | |
| Pescara del Tronto | Photos | Nadir | 4000x3000p (12 MP) **shutter speed = 1/60 sec** | 166 | 5.91 | Between the 5 and 9-09-16 |
| | Video | Oblique | 1920x1080p (2.1 MP) | 18 | **3.8** | 26-8-16 |
| | GCPs | 25 GCPs acquired from well distributed marks, such as building corners in Pescara del tronto. | | | | |

**Table 1.** Data obtained at the study sites. Highlighted are some data propieties that are especially relevant for this research.

## 2.2 Data preparation

Several datasets were generated from the acquired video-streams and photos in relation to the following preparation parameters to be investigated: (i) data type, (ii) frame selection approach, (iii) sensor perspective and (iv) resolution. (i) First datasets were divided according to their nature into video footages and photos for every study case. (ii) Then video frames were extracted by two approaches: Random Frame Selection (RFS) and a Wise Frame Selection approach (WFS). RFS is only defined by an empirical number of randomly-selected frames, while WFS refers to a more elaborated approach that reduces blur-motion effects and frame redundancy. The latter uses as guideline their initial 3D position generated with Pix4D software (Pix4D, 2017), with the aim to avoid redundancy, complemented with an Image Quality Index (IQI) computed for every frame using 3DFlow software (3DFlow, 2017), to ensure the selection of frames with the highest quality. (iii) Additionally, the Tainan photos were divided according to their perspective into oblique and nadir. Video frames by their nature can be considered oblique. (iv) Finally, extracted video frames and photos resolutions were modified to generate additional datasets; four levels were analyzed: original, medium, low and coarse (i.e. very low) resolution. Twenty-six datasets were prepared in total, each related to a different preparation parameter to be investigated, as summarized in Table 2. Of those, two video and two photo datasets were labeled as default, since they were prepared with default parameters (i.e. Random Frame Selection in case of videos, oblique and nadir perspective in case of photos, original resolution and without refinement). Then, 3DPCs were generated from the generated datasets with Pix4d. One reference 3DPC was prepared using a refinement procedure which





consisted on the use of Manual Tie Points (MTP) in Pix4d. Finally, the GCPs were used for the geo-registration of most photo-based 3DPCs, whereas the video and Tainan oblique photos-based 3DPCs were registered using corresponding extracted 3D points from the reference 3DPC.

| No. | Study site | Data type | Frame selection approach | Sensor perspective | Resolution* | Refinement | Date | Geo-registration |
|-----|------------|-----------|--------------------------|--------------------|-------------|------------|------|------------------|
| 1 | Tainan | Photo | N/A | Oblique and nadir | Original | No | 07/02/16 | GCPs |
| 2 | Tainan | Photo | N/A | Oblique and nadir | Original | Yes | 07/02/16 | GCPs |
| 3 | Tainan | Photo | N/A | Oblique and nadir | Medium | No | 07/02/16 | GCPs |
| 4 | Tainan | Photo | N/A | Oblique and nadir | Low | No | 07/02/16 | GCPs |
| 5 | Tainan | Photo | N/A | Oblique and nadir | Coarse | No | 07/02/16 | GCPs |
| 6 | Tainan | Photo | N/A | Oblique | Original | No | 07/02/16 | GCPs and reference 3DPC |
| 7 | Tainan | Photo | N/A | Nadir | Original | No | 07/02/16 | GCPs |
| 8 | Tainan | Photo | N/A | Oblique and nadir | Original | No | 10/02/16 | GCPs |
| 9 | Tainan | Photo | N/A | Oblique and nadir | Original | No | 11/02/16 | GCPs |
| 10 | Tainan | Photo | N/A | Oblique and nadir | Original | No | 12/02/16 | GCPs |
| 11 | Tainan | Photo | N/A | Oblique and nadir | Original | No | 13/02/16 | GCPs |
| 12 | Tainan | Photo | N/A | Oblique and nadir | Original | No | 14/02/16 | GCPs |
| 13 | Tainan | Video frame | RFS | Oblique | Original | No | 7/2/16 | Reference 3DPC |
| 14 | Tainan | Video frame | RFS | Oblique | Medium | No | 7/2/16 | Reference 3DPC |
| 15 | Tainan | Video frame | RFS | Oblique | Low | No | 7/2/16 | Reference 3DPC |
| 16 | Tainan | Video frame | RFS | Oblique | Coarse | No | 7/2/16 | Reference 3DPC |
| 17 | Tainan | Video frame | WFS | Oblique | Original | No | 7/2/16 | Reference 3DPC |
| 18 | Pescara del Tronto | Photo | N/A | Nadir | Original | No | 5-9/9/16 | GCPs |
| 19 | Pescara del Tronto | Photo | N/A | Nadir | Medium | No | 5-9/9/16 | GCPs |
| 20 | Pescara del Tronto | Photo | N/A | Nadir | Low | No | 5-9/9/16 | GCPs |
| 21 | Pescara del Tronto | Photo | N/A | Nadir | Coarse | No | 5-9/9/16 | GCPs |
| 22 | Pescara del Tronto | Video frame | RFS | Oblique | Original | No | 26/8/16 | Reference 3DPC |
| 23 | Pescara del Tronto | Video frame | RFS | Oblique | Medium | No | 26/8/16 | Reference 3DPC |
| 24 | Pescara del Tronto | Video frame | RFS | Oblique | Low | No | 26/8/16 | Reference 3DPC |
| 25 | Pescara del Tronto | Video frame | RFS | Oblique | Coarse | No | 26/8/16 | Reference 3DPC |
| 26 | Pescara del Tronto | Video frame | WFS | Oblique | Original | No | 26/8/16 | Reference 3DPC |

Dataset 1 and 18: Default photo datasets. The dataset 1 was also used for the multi-temporal application analysis together with datasets 8 to 12.

Dataset 13 and 22: Default video datasets. Used also for the analysis of frame selection approach as RFS in comparison to WFS (datasets 17 and 26).

Dataset 2: Dataset used as reference 3DPC. This is the only dataset that followed a refinement process with Manual Tie Points.

*Original resolution: 4000x3000p or 12 MP (Photos) and 1920x1080p or 2.1 MP (Video frames), Medium resolution: 1/2 (Half image size), Low resolution: 1/4 (Quarter image size) and Coarse resolution: 1/8 (Eighth image size)

**Table 2.** Prepared datasets from video and photo data



### 2.3 Usability of video data for 3DSDA

Once all the datasets and 3DPCs were prepared, as described in Subsection 2.2, their usability was analyzed in three sections:

**2D Quality assessment** Consisted on a direct measurement of the image quality of every dataset with the use of IQI, complemented by the assessment of their depictability of 2D damage-related features for 2D-based SDA.

5 **3D Quality assessment** Consisted on the determination of the absolute geometric quality of the generated 3DPCs expressed by their internal and external accuracy, complemented by their depictability of 3D damage-related features for 3D-based SDA.

**Application analysis** Based on the estimation of the debris volumes with the generated 3DPCs, determined by the analysis of the debris change trend and accuracy assessment.

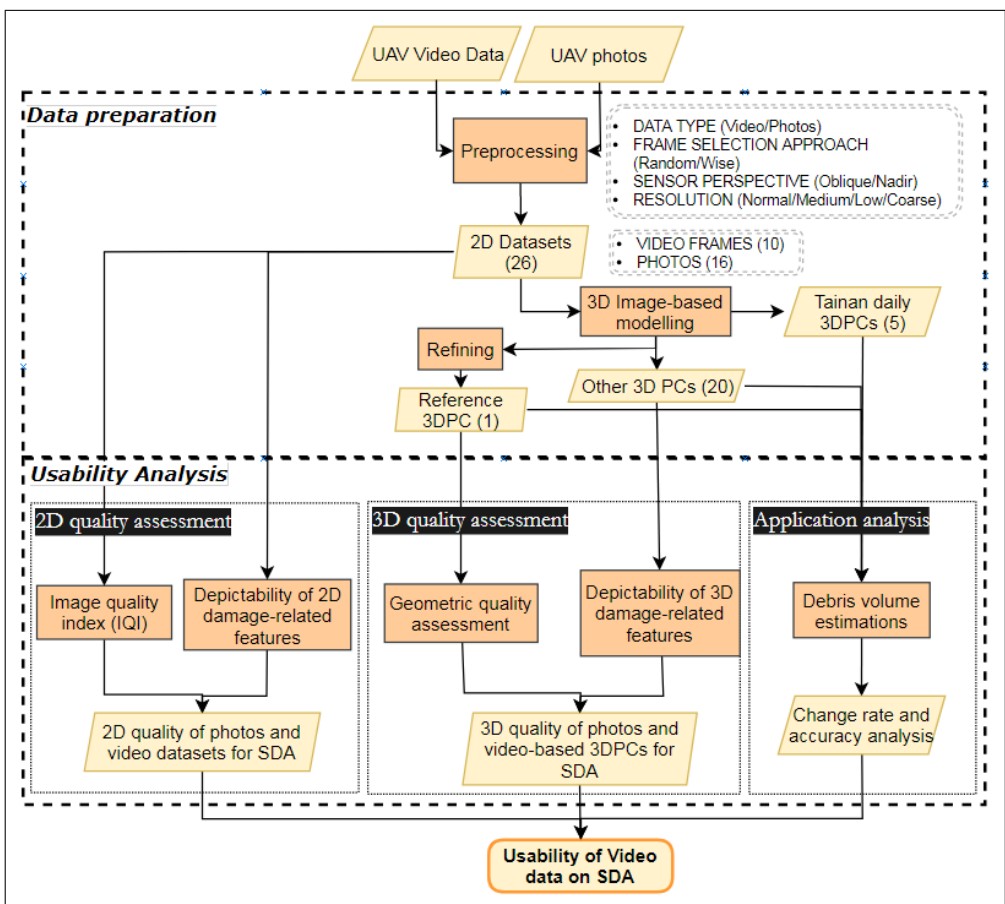

**Figure 2.** Workflow to determine usability of video data in 2D and 3D-based SDA. Dashed rectangles: Main phases.



### 2.3.1 2D Quality Assessment

*Direct image quality assessment with Image Quality Index (IQI)*

The prepared datasets were first analyzed with their respective IQIs for their direct quality assessment. IQI series and image reference quality maps indicate possible sources of error for image-based 3D scene reconstruction such as the presence of

low-texture areas or motion-blur effects (3D Flow, 2017).

*Depictability of 2D damage-related features*

This assessment was based on the classification accuracy of damaged and non-damaged segments or super-pixels extracted from the prepared datasets. A deep learning approach presented by Vetrivel et al. (2017) was tested, this uses the 'imagenet-caffe-alex' (Krizhevsky et al., 2012) CNN model as extraction tool of damage-related features. Three representative video

frames and photos of the same area were chosen from both data types and study areas based on the damage presence: (i) severe/complete damage, (ii) partial damage and (ii) no damage. Those datasets were first segmented into super-pixels using the Simple Linear Iterative Clustering (SLIC) algorithm (Achanta et al., 2010). Empirical values were tested according to the data scale with the aim of over-segmenting the scene. The extracted super-pixels were then classified into damage and non-damage categories using the CNN model. Damage classification accuracies for every dataset were determined using reference

classifications created by visual inspection of the same segmented datasets. Finally, the depictability of damage-related features for the analyzed datasets was determined by comparing their damage classification accuracies.

### 2.3.2 3D Quality Assessment

*Geometric quality assessment*

Point cloud geometric quality was determined by a 3DPC internal and external accuracy assessment. Internal accuracy is related

to the model precision and different indicators can measure this 3DPC attribute. Based on previous methodologies (Oude Elberink and Vosselman, 2011; Jarzabek-Rychard and Karpina, 2016; Soudarissanane et al., 2008), in this research planar fitting and completeness of the generated 3DPCs were measured. For the former a planar object (e.g. roofs) was identified and extracted as 3DPC segment from every 3DPC and study area using Cloud Compare (CloudCompare, 2017). These segments were used to define a plane by a least squares fitting algorithm. The mean distance of every 3D point within the segment to

its fitted plane was measured to determine an absolute deviation for every dataset. To mitigate the effect of flying height, a relative deviation was measured by computing the ratio of the obtained mean distances and the dataset respective GSD (Table 1). Completeness, in turn, was measured from the projection of every 3DPC to two raster maps, each associated with a different perspective (i.e. vertical and horizontal)(Girardeau-Montaut, 2017). Raster cells in these maps containing the number of 3D points were used to compute mean density and the percentage of empty cells for every 3DPC.

External accuracy instead refers to how approximate a 3DPC is to reality, which requires a reference model (Alsadik et al., 2015; Kersten and Lindstaedt, 2012). The reference 3DPC (Table 2: Dataset 2; see Subsection 2.2) was used and compared to every 3DPC. Thereby, the mean distance of the 3D points to the reference 3D points was computed to determine every 3DPC external accuracy.



*Depictability of 3D damage-related features*

This assessment was based on previous investigations on the use of 3DPCs for SDA. Visual inspection was applied to 3DPCs for the identification of severe-damaged buildings in Fernandez Galarreta et al. (2015). In a similar manner, here different damage-related features where identified using a mesh model generated from the studied 3DPCs. Thus, the feasibility on this

visual identification was compared and interpreted. As complement, a more elaborated approach using 3D damage-related features based on Vetrivel et al. (2017) was applied. This consisted of a segment-level representation of different 3D features. Three 3D features were used: Mean Curvature, Normal Change Rate and Roughness. These point-level features were estimated using a kernel (neighborhood size) of 1 m for all 3DPCs. A damaged area in Tainan was delineated to retrieve the segment-level distributions of these 3D features for every 3DPC, including the reference 3DPC (Table 2: Dataset 2). Through the use of box-

plots, the segment-level distributions derived from every photo and video-based 3DPC were compared to the ones retrieved from the reference model to determine their depictability of 3D damage-related features. Additionally, non-damage segment distributions were retrieved from the reference 3DPC to confirm the relation of the chosen 3D features with damage (Figure 3).

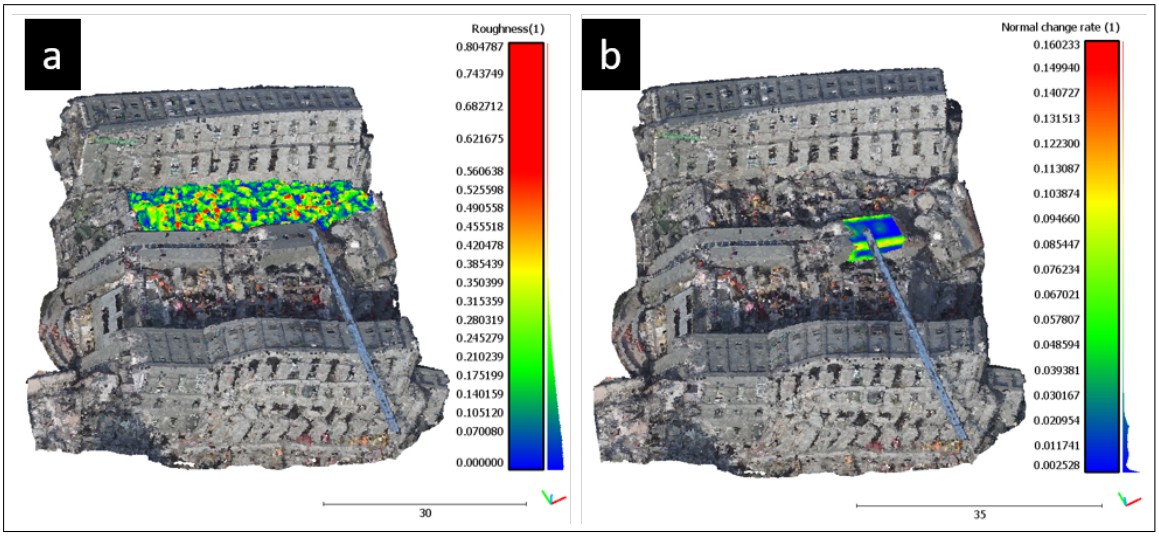

**Figure 3.** Segments selected for the depictability of damage-related features analysis. (a) Damage segment (Roughness values) (b) Non-Damage segment (Normal Change Rate values)

### 2.3.3    Application analysis

UAV photos of six different dates after the Tainan earthquake on 6 February, 2016, were used in this analysis: 7, 10, 11, 12, 13

and 14 February, 2016. The 3DPC for the first date was generated with the Tainan photos with default preparation parameters (Table 2: Dataset 1), as were the 3DPCs for the subsequent dates (see Subsection 2.2; Table 2: Datasets 8 to 12). From the 3DPC of the first date the damaged area was delimited and used as base value for the volume estimations. The volume of debris presented within this area was calculated in $m^3$ and analyzed for every date using Pix4d. The same volume estimation approach was applied to the 3DPC generated with the video default dataset (Table 2: Dataset 13) and the reference 3DPc (Table



2: Dataset 2). Thereby, volume estimations with photo-based 3DPC were analyzed and later compared to the video-based and reference estimations to determining their similarity and accuracy.

## 3 Results

### 3.1 2D Quality Assessment

*Direct image quality assessment with Image Quality Index (IQI)*

Lower IQI values for video datasets were observed in the Tainan case, compared to all photo datasets (Figure 4). The WFS dataset have greatly reduced variance and in the case of Pescara del Tronto raised IQI values. Of the oblique and nadir photos, the nadir dataset showed less variance but similar IQI values in the case of Tainan. In Pescara del Tronto the nadir dataset contained substantial variance. A direct relation between IQI values with resolution was observed also when analyzing resolution-

degraded datasets, the better the resolution the higher IQI values.

*Depictability of 2D damage-related features*

Similar extraction and classification accuracies were obtained for both data types (Table 3). Between the two study areas, higher accuracies of 84 and 82% were achieved for the Pescara del Tronto video and photo datasets, respectively. For Tainan they were of 77 and 78%. Errors were mainly associated with false positives (i.e. misclassified and extracted damage features),

caused by texture-rich super-pixels with the presence of cars or people in Tainan, and low texture super-pixels, such as bare ground, in the case of Pescara del Tronto (Figure 5).

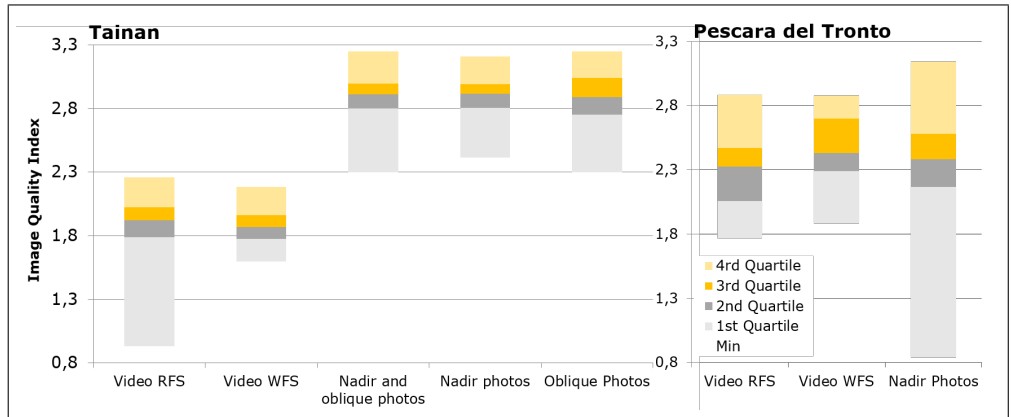

**Figure 4.** Boxplot of the Image Quality Index (IQI) computed for Tainan and Pescara del Tronto datasets.

### 3.2 3D Quality Assessment

*Geometric quality assessment*

Planar fitting and completeness assessments were performed to determine internal accuracy. For the former large deviations



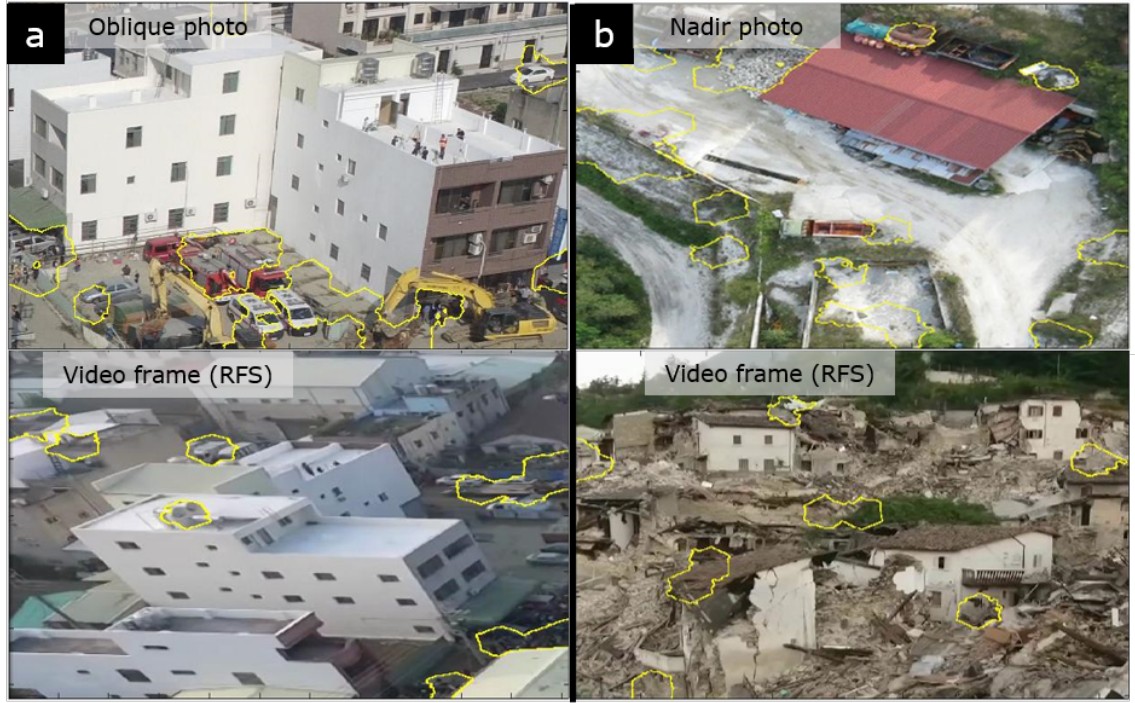

**Figure 5.** Extraction of 2D damage-related features using different datasets. Polygons: False positives in the extraction of 2D damage-related features for (a) Tainan oblique photo and video frame (Random Frame Selection RFS) and (b) Pescara del Tronto nadir photo and video frame (RFS).

| Tested Data | Tainan | | Pescara del Tronto | |
|---|---|---|---|---|
| | **Video** | **Photo** | **Video** | **Photo** |
| Severe damage | **73** | 67 | 72 | **81** |
| Partial damage | 71 | **72** | **90** | 74 |
| Non-damage | 88 | **94** | 87 | **90** |
| Mean Accuracy | 77 | 78 | 84 | 82 |

**Table 3.** Accuracy (%) of 2D damage-related feature extraction for every dataset and study case.

from the chosen planar object, and thus low precision, were computed in the case of Tainan for the 3DPCs generated with the combined oblique and nadir photos (Figure 6). The software used was not able to merge oblique and nadir photos resulting in a highly displaced 3DPC. Improved precision was obtained when using individually the oblique or nadir photo datasets; however in the Tainan case, RFS video-based 3DPC is still more precise, or at least presents less relative 3DPC deviation than the oblique
5  photo-based 3DPC. The 3DPCs generated with nadir photos presented the lowest deviation and were the most precise in both study cases. The 3DPC generated with the WFS video dataset presented more deviation than the one generated with the RFS





video dataset in both study cases. Contrasting effects of resolution were found between the study cases. It was observed that resolution degradation can improve precision by reducing 3DPC noise in the Tainan case, while the opposite effect occurred in Pescara del Tronto from medium resolution downwards. Besides, the completeness analysis demonstrated that the 3DPCs generated with oblique photos were denser than the video-based 3DPCs in the Tainan case. The RFS video-based 3DPC in turn

5   was denser than the one generated from nadir photos in both study cases. However, considering the percentage of empty cell, the RFS video-based 3DPCs percentages are low, and outperform percentages of oblique and nadir-based 3DPC in the vertical and horizontal perspective, respectively (Figure 7 and Table 4). The WFS video dataset produced low-density 3DPCs with a large percentages of empty cell in both study cases and perspectives.

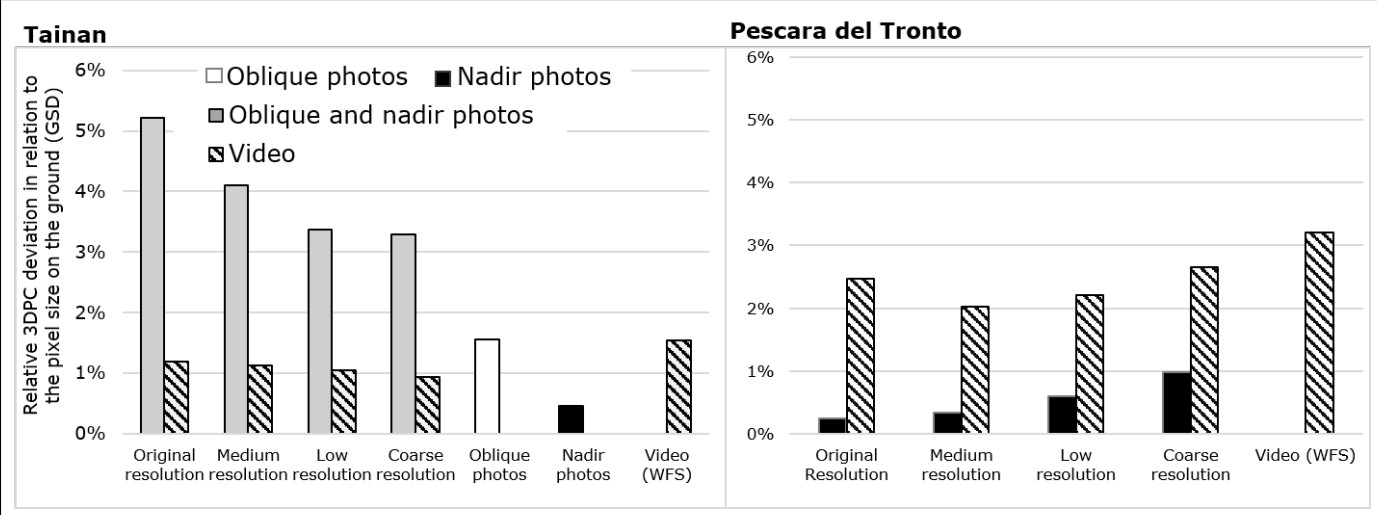

**Figure 6.** Planar fitting analysis of Tainan and Pescara del Tronto 3D Point Clouds as part of the internal 3D quality assessment. Comparison of the mean distance of 3D points to the theoretical planar object. Larger distance implies lower 3D point cloud precision.

Concerning external accuracy, the combined oblique and nadir photos-based 3DPC also resulted highly inaccurate. Indepen-

10   dent oblique and nadir photos-based 3DPCs registered higher accuracies of 2 and 4 cm respectively (Figure 8). Video-based 3DPCs, with an accuracy of 7 cm, resulted less accurate than individual oblique or nadir photos-based 3DPCs. It was observed that the WFS video-based 3DPCs removed some noise in 3DPCs (i.e. less variability), but its accuracy was still the same of RFS video-based 3DPCs.



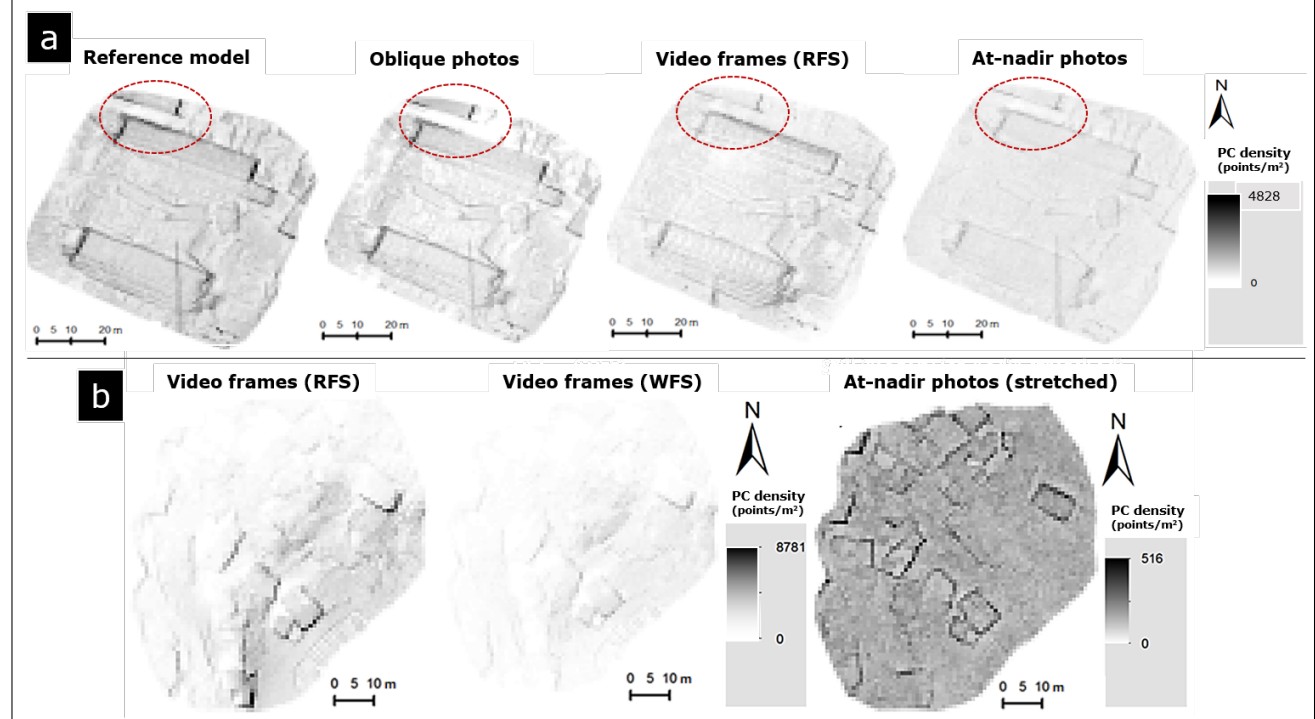

**Figure 7.** Vertical point cloud density raster maps generated for the completeness analysis. (a)3D Point Clouds of Tainan and (b) of Pescara del Tronto. Dashed red circle: noticeable difference in completeness between 3DPCs of Tainan.

| Study case | Perspective | Reference 3DPC | 3D Point Cloud (PC) source | | | |
|---|---|---|---|---|---|---|
| | | | Nadir photos | Oblique photos | Video (RFS) | Video (WFS) |
| **Tainan** | Vertical | 779 | 408 | **519** | 438 | 135 |
| | | 0% | **0%** | 4% | **0%** | 2% |
| | Horizontal | 3129 | 1649 | **1967** | 1816 | 573 |
| | | 0% | 3% | **2%** | 3% | 3% |
| **Pescara del Tronto** | Vertical | | 167 | | **667** | 255 |
| | | | **0%** | | 4% | 8% |
| | Horizontal | | 499 | | **1752** | 667 |
| | | | 4% | | **1%** | 3% |

**Table 4.** Computed mean point cloud densities (points per cell or m$^2$) and proportion of empty cells (%) from horizontal and vertically projected point cloud density raster maps.





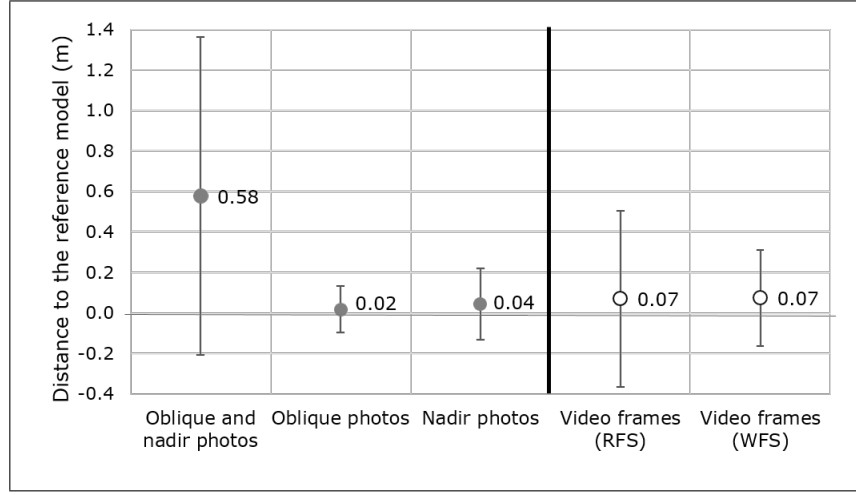

**Figure 8.** External accuracy assessment of video and photo-based 3DPCs of Tainan. Comparison of the mean distance of 3D points between the analyzed generated and reference 3D Point Clouds.

### Depictability of 3D damage-related features

In the case of Pescara del Tronto building deformations, cracks and debris features could be readily identified by visual inspection in video-based mesh models. These models outperformed representations of nadir photos-based 3DPC and even better representations were obtained using degraded-resolution video frames (Figure 9). In the Tainan case, cracks and spalling fea-

tures were better represented in the oblique photos-based models than in the video-based ones; however, their identification was still difficult specially compared to the reference mesh. Other damage characteristics, such as inclined walls and spalling were only extractable from the reference model.

The analysis of the distribution of 3D damage-related features showed more deviating distributions, therefore less depictability in photo-based 3DPCs (Figure 10). RFS video-based 3DPCs distributions were consistently aligned with the distributions

extracted from the reference 3DPC for all 3D features analyzed. The distributions of the other 3DPCs showed deviations in at least one feature. The relation of these features with damage characteristics was confirmed by the inclusion and analysis of a non-damaged segment extracted from the reference 3DPC, which clearly presented deviations in all 3D features.

### 3.3 Application analysis

Debris volume change was analyzed for Tainan using multi-temporal photo-based 3DPCs, and as expected a negative trend was

observed in accordance with the cleanup operations (Figure 11). However, the comparison with the estimation made with the reference 3DPC for the first date (7 February, 2016), revealed a large overestimations in these series. Debris volume estimation made with the 3DCP generated from the default dataset (i.e. the combined oblique and nadir photos) was of around 8000 $m^3$ more than the estimation done with the reference 3DPC. This is linked to the displacement effects observed in the 3D external accuracy assessment. In contrast, volume estimations made with the oblique photo-based 3DPCs and video RFS dataset for the



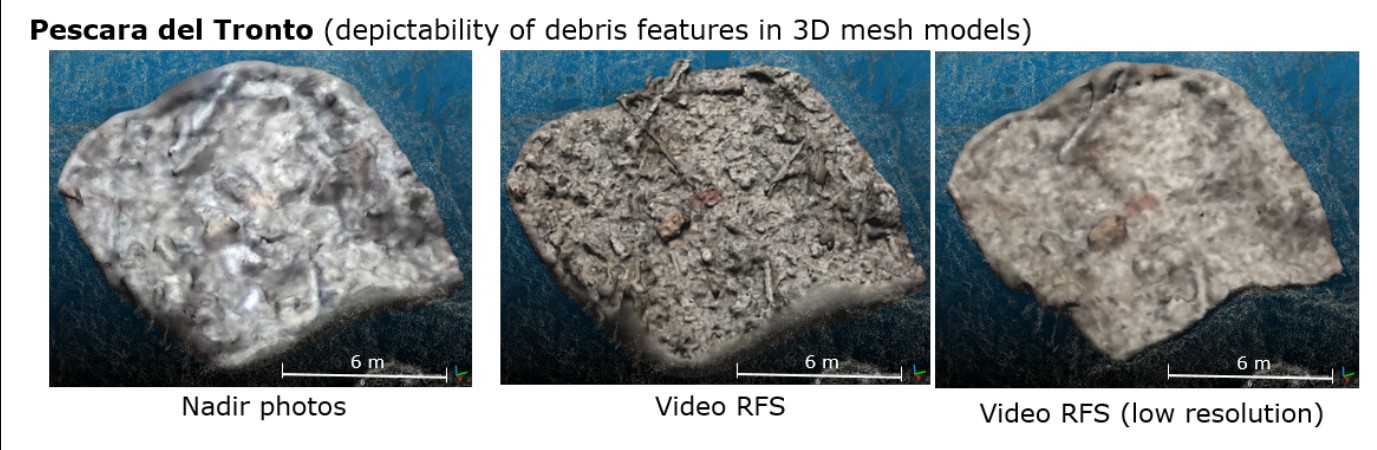

**Figure 9.** Comparison of the depictability of debris features in photo and video-based mesh models of Pescara del Tronto. Lower depictability in nadir photos-based mesh due to their high GSD and the presence of motion-blur effects.

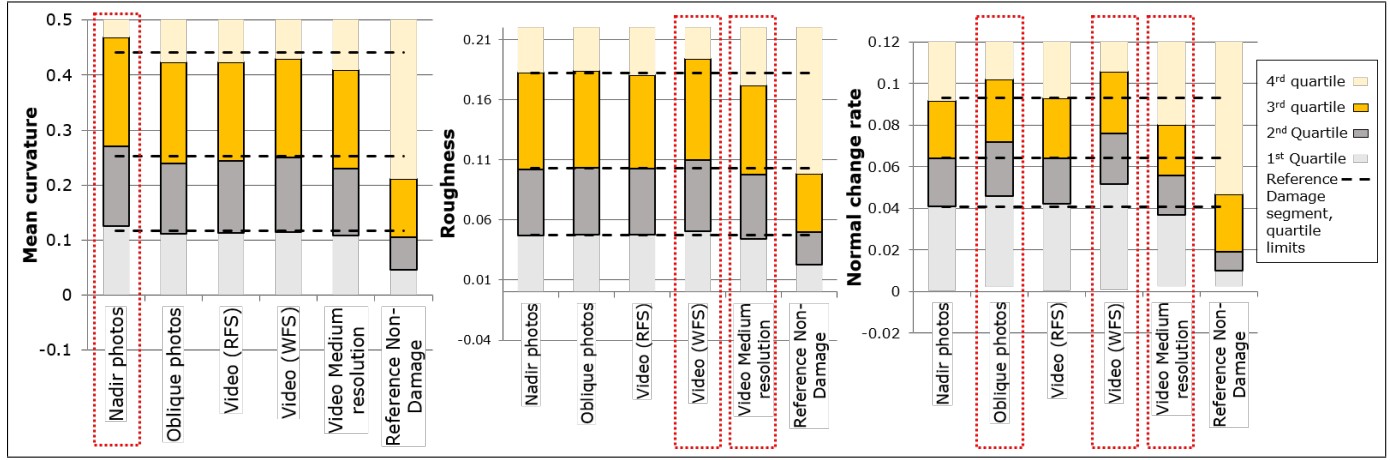

**Figure 10.** Tainan segment-level distributions of 3D damage-related features. Doted rectangles highlight discordant distributions from the reference damage segment and thus low depictability of 3D damage-related features.

same date presented more accurate results. The estimation with the oblique-based 3DPC was only 486 m$^3$ above the reference (1.6% of error); while, the most accurate estimation was obtained with the RFS video-based 3DPC, with an error of only 136 m$^3$ (0.4%).

## 4    Discussions

5    In the 2D quality analysis lower IQI values for the video datasets were obtained in comparison to photos in the case of Tainan, while they were similar in the case of Pescara del Tronto. Low video IQI values were caused by low resolution and smoke





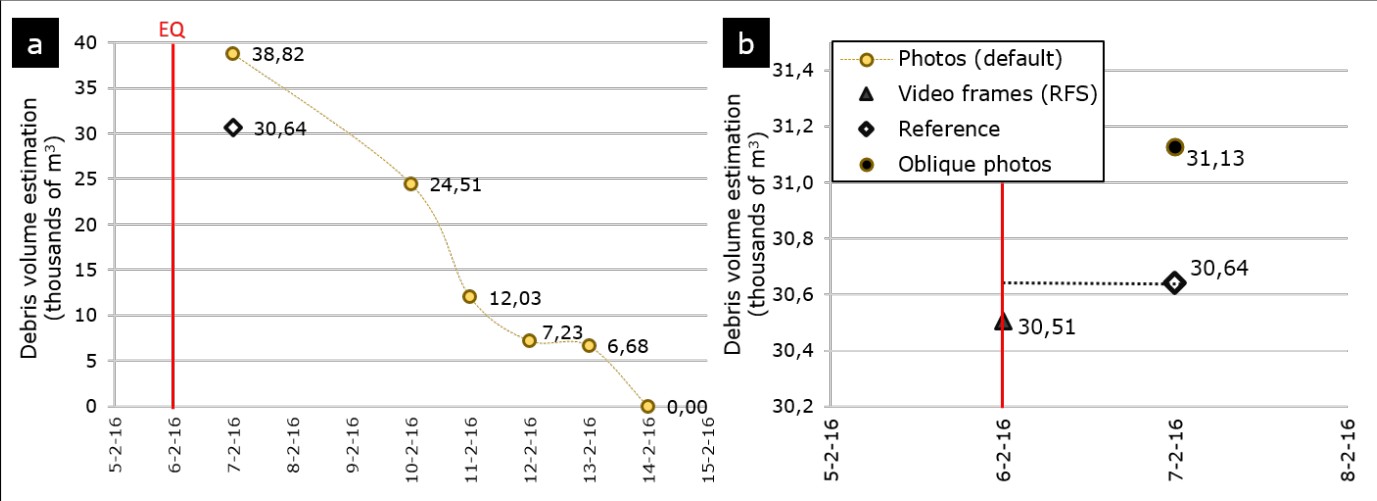

**Figure 11.** (a) Debris volume change analysis using the Tainan multi-temporal photo-based 3DPC and reference model. (b) Volume estimations for video-based 3DPC (Random Frame selection), oblique photos-based 3DPC and reference model. Red line: Earthquake (EQ) date.

presence in Tainan video frames, whereas in the case of Pescara del Tronto these were comparable since at-nadir photos were affected by a large GSD and the presence of motion-blur effects, generated during the acquisition of nadir photos by the high altitude and low shutter speed, respectively (see Subsection 2.1). The WFS dataset improved IQI values and reduced variance in comparison of RFS video dataset, since WFS approach filtered frames affected by external factors (e.g. texture-low

zones, smoke presence and moving objects), clearly indicated by low IQI values in IQI series (Figure 12). Despite direct video 2D quality was lower than the one of photos, similar accuracies were obtained from the depictability of 2D damage-related analysis. In this analysis, also external factors were responsible for false positive errors and low accuracy in the classification of 2D damage-related features, rather than the video data quality characteristics. Moreover, the analyzed video frames were chosen from the RFS dataset; therefore, higher accuracies could be obtained using frames from the WFS dataset. Low resolution was

the only video characteristic that affected 2D quality, other data-related quality artifacts, such motion-blur effects were instead identified in nadir photos. Therefore, sufficient 2D quality and accurate depictability of damage-related features of video data, demonstrated that they can perform 2D-based SDA as accurate as photos.

     Concerning the 3D quality assessment, comparable qualities were obtained for video and photo-based 3DPCs. Photo-based 3DPCs of Tainan generated with the combined oblique and nadir dataset and its degraded-resolution datasets ((Table 2: Datasets

1 to 5), produced largely displaced 3DPCs that presented the lowest quality (Figure 13). A main possible cause is the large difference in the spatial and spectral characteristics of the combined datasets which affected the Automatic Tie Point selection during the initial 3D modeling step. A less probable cause is a mis-registration error due to the lack of GCPs for orienting oblique photos and their low z-variability (see Subsection 2.2 and 2.1). A solution would be a refinement procedure; however, this implies a time-demanding operation that hinders photo data usability. In contrast, 3DPCs generated independently from




nadir and oblique photos showed higher qualities. 3D qualities of video-based 3DPCs were also high and comparable to the ones of oblique or nadir photos-based 3DPCs. In some quality measurements, video-based 3DPCs even outperformed the quality of the 3DPCs generated from oblique or nadir photos. For example, the Tainan RFS video-based 3DPC was denser and presented a lower percentage of empty cells; occluded areas were uniquely identified in the RFS video-based 3DPC (Figure

7). The better performance of video data in 3DPC generation is associated to their small GSD which compensated their lower resolution, their complete scope of the scene in the case of Tainan, and the motion-blur that affected the nadir photos-based 3DPCs in the case of Pescara del Tronto. Similarly, in the depictability of 3D damage-related features analysis, 3D damage-related features were in many cases better identifiable in the RFS video-based 3DPC than in the nadir and oblique photo-based 3DPCs, both visually and using the CNN model. The influence of video artifacts and quality characteristics was not evident

either in this quality analysis. Most effects were produced by external factors, such as the presence of smoke and changes in the scene in the case of Tainan, and hilly topography which affected completeness in the case of Pescara del Tronto. However, the influence of external factors was still marginal compared to the favorable small GSD and coverage of video frames. In relation to the WFS approach, 3DPC noise mainly related to smoke in Tainan was reduced by filtering the affected frames; however it did not improve, and even reduced the quality obtained using the RFS video-based 3DPC. The limited number of video frames

in the WFS dataset resulted in low 3DPC precision, completeness and depictability of 3D damage-related features. Noise in 3DPCs was also reduced when using the down-sampled Tainan video and photo datasets. Proximate or even higher 3D quality parameters of RFS video-based 3DPCs, together with their proper depictability of 3D damage-related features, indicated that also video frames can be used for accurate 3D-based SDA.

The application analysis lead to results consistent with the 3D quality assessment, since also here the displacement effect of

the combined oblique and nadir photo-based 3DPC produced a large over-estimation in comparison with the reference. Video data in spite of their low resolution and the influence of all the mentioned external factors achieved an accurate estimation, comparable to the one obtained the oblique photo-based 3DPC. Thereby, it was demonstrated that the RFS video-based 3DPC was able to perform debris volume estimation application with the same or better accuracy than with the photo-based 3DPC.

Results obtained in this research approach the ones observed in previous studies. Accuracies achieved for the depictability of

2D damage-related features analysis are similar to the ones obtained in Vetrivel et al. (2017), mainly for the case of Pescara del Tronto. Likewise, a similar external accuracy of 5 cm was obtained by Alsadik et al. (2015) for video-based 3DPCs, 2cm more than the one obtained in this research. Here the method used to select the highest-quality video frames (i.e. WFS) produced low-density point clouds, in contrast to Xu et al. (2016) who claimed that the reduction of motion-blur frames generated denser 3DPCs. The authors, however, did not consider redundancy whose reduction implies the elimination of a large number of

frames. Dealing with frame redundancy remains a challenging issue, until now empirical values were used to define a proper frame selection rate, though this is data and case-specific (Alsadik et al., 2015; Clift and Clark, 2016). A video characteristic not considered but that influenced the accuracy of the produced 3DPC is the lack of proper geo-positional information. All video-based 3DPCs were registered in relation to the reference 3DPC; however, the lack of reference information or adequate GCPs will affect usability of this kind of data.




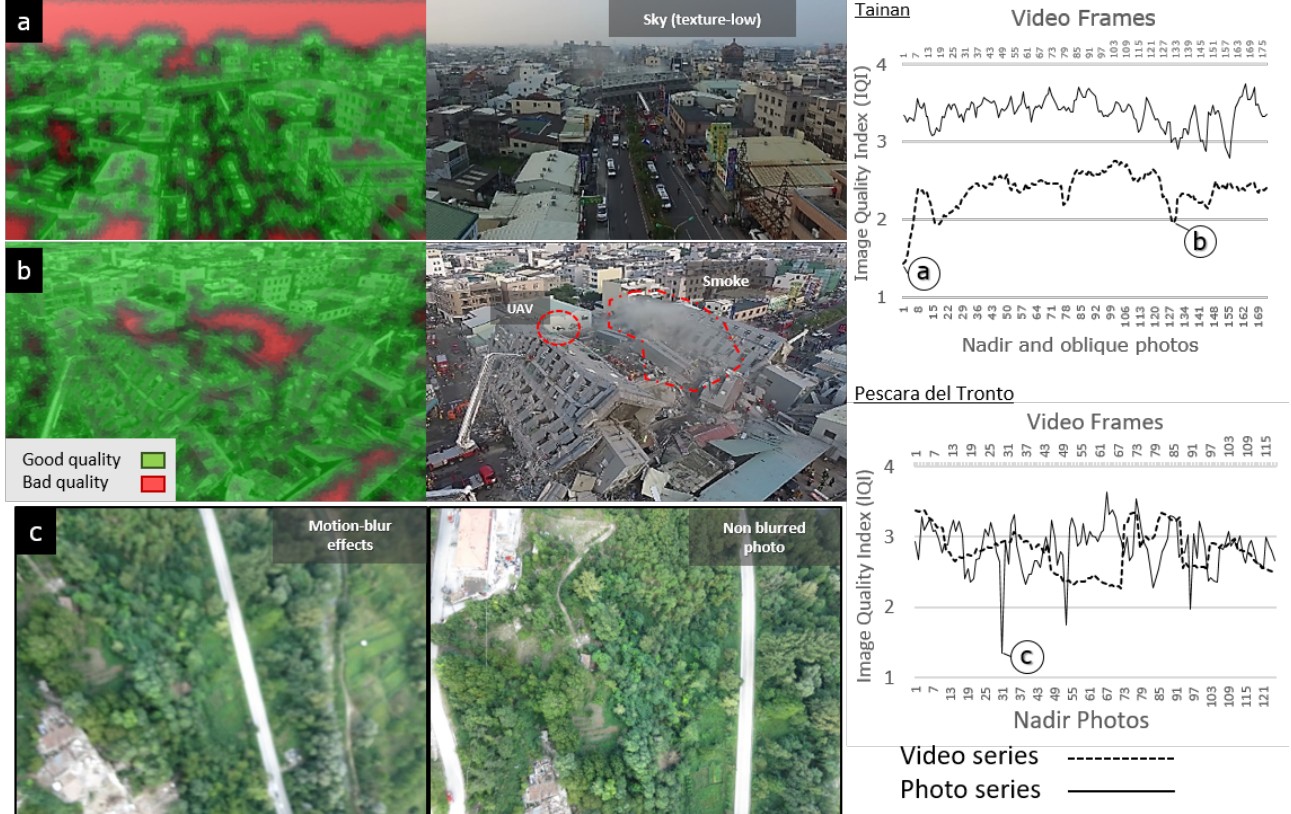

**Figure 12.** Image Quality Index (IQI) series for Pescara del Tronto and Tainan. (a) The sky as a low-texture area is considered a likely source of 3D modelling error in IQI maps (red area). (b) Also smoke and moving objects (UAV). (c) In Pescara del Tronto motion-blur effects result in IQI values.

## 5 Conclusions

The usability of video data for SDA was investigated based on the analysis of their 2D and 3D quality and application in post-disaster activities. The 2D quality analysis indicated possible sources of error expressed by low IQIs for the video datasets, attributed to low resolution and external factors (e.g. smoke presence, low-texture zones). However, the analysis of the de-

5    pictability of 2D damage-related features demonstrated that this kind of data can be used for an accurate 2D-based SDA. Video-based 3DPCs in turn exhibited a geometrical quality comparable to 3DPCs generated with photos acquired with still cameras. The 3DPCs generated with the RFS video dataset were in certain cases more precise and complete than those from nadir or oblique photos. External accuracy identified slightly less accurate 3DPCs generated with the RFS video dataset; however, based on the depictability of 3D damage-related features analysis, it is sufficient for 3D-based SDA. Similarly, the RFS

10   video-based 3DPC was able to estimate more accurately debris volume than models based on oblique photos, confirming the good usability of video data. Data resolution was the main quality parameter affecting video 2D and 3D quality; however,





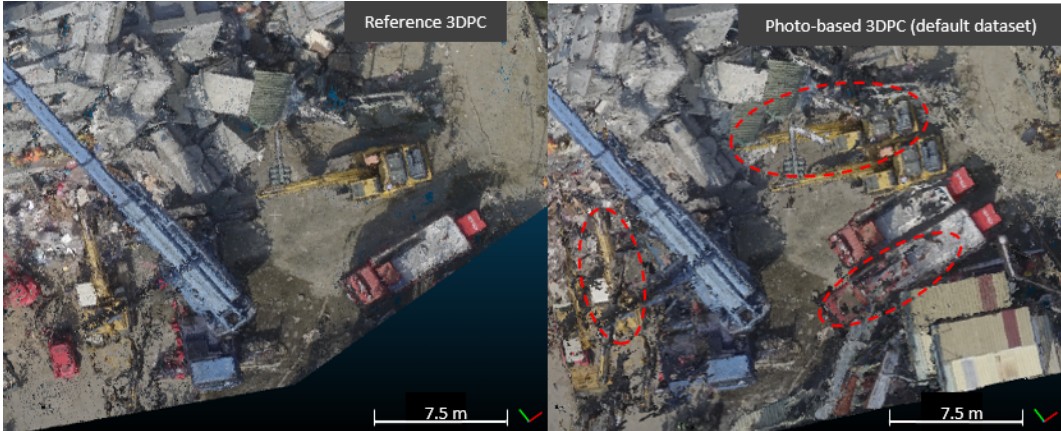

**Figure 13.** Displacement effect in 3DPC generated with the default photo dataset (combined nadir and oblique photos), due to a misregistration or differences in spectral and spatial characteristics between oblique and nadir datasets. Red dashed circles: noise in 3DPCs.

it has been seen that a small GSD (i.e. low altitude) complemented with a complete scope of the scene can compensate low video resolution effects. Consequences of other data artifact and quality characteristics, such as redundancy and motion-blur effects were not identified, and in contrast, external factors such as smoke presence, terrain texture or changes in the scene were more influencing on video data quality and usability. Moreover, the reduction of redundant video frames can also result in low

quality 3DPCs, as observed with WFS approach. A relevant quality parameter not considered in this research is the availability of accurate GCPs for 3DPC geo-registration. The absence of proper GCPs, mainly for oblique images or video frames, may affect generally the 3D quality, as the displacement observed in the 3DPC generated using the combined oblique-nadir photos. Methods for direct geo-registration, such as described by Turner et al. (2014), could be applied and tested with video data. Also, as video data were limited to 2.1 MP resolution, better performance is expected for current 4K Ultra High Definition

(UHD) videos of 8.8 MP. Both aspects could be investigated by a synthetic experiment as continuation of this research, and this may also contribute in the application of video data for real-time damage estimations with deep learning methods.

*Competing interests.* The authors declare no conflict of interest.

*Acknowledgements.* The work was supported by FP7 project INACHUS (Technological and Methodological Solutions for Integrated Wide Area Situation Awareness and Survivor Localisation to Support Search and Rescue Teams), grant number: 607522. The data used in this

research was acquired and kindly provided by: Jyun-Ping Jhan, Institute of Photogrammetry-University of Stuttgart and Jiann-Yeou Rau, National Cheng Kung University, Tainan (Tainan UAV photos and GCPs), and Filiberto Chiabrando Politecnico di Torino, Italy (Pescara del Tronto UAV photos and GCPs). The CNN damage classification model was in turn provided by Anand Vetrivel who also contributed to this research with insight and expertise.





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
