# Peer review of "Usability of aerial video footage for 3D-scene reconstruction and structural damage assessment"

_Natural Hazards and Earth System Sciences, 2017_

## Referee Comment (RC1) · Anonymous Referee #1 · 23 Feb 2018

The paper is well written and interesting, as it goes through and evaluates different approaches for automated recognition of damage-related features based on rapidly available imagery datasets, for an effective post-disaster structural damage assessment.

The topic of the paper is interesting since it explores the potentialities of using rapidly collectable image datasets provided by video surveys, that are normally performed immediately after the occurrence of disasters in built-up areas. The difference between the results obtained using conventional photo dataset (nadir and oblique) and those obtained using video frames, along with advantages and drawbacks of each approach are clearly explained and supported by valid and exhaustive statistics. This research can represent a good starting point for further studies mainly focused on the optimized

use of video frames not only for rapid SDA but, with the advent of new 4K (or more) resolution video cameras, even for nearly-real time ground modeling. However, there are some minor corrections that I can suggest to the authors, to make the paper possibly more clear and self-explanatory.

The results are sufficient to support the interpretations and the conclusions, nevertheless a few points need to be further clarified: 1) Probably reducing the frame redundancy using the WFS for video data can generate a lack of important photos, needed for an accurate Point Cloud calculation. For that reason, the video results with WFS are generally worse than the RFS. Maybe the WFS method is not appropriate in this case because the photos selected as good were in a non-optimal position. Is clear that the results of a Point Cloud generation are strongly related to the overlap and sidelap percentage between photograms. Maybe with different rules for WFS the results could be even better than using the RFS, even if the latter method is certainly faster. 2) Considering 3D Internal Accuracy assessment. The authors use planar fitting as a method to assess the accuracy of the resulting 3DPCs. It seems that only one planar object was used to perform this analysis. Maybe, considering more planar objects preferably equally distributed, for example, with respect to shaded and well-lighted areas or central or peripheral position could lead to more robust conclusions. 3) Considering 3D External Accuracy: it is not clear how the distance between each 3DPC and the reference one is considered. Is it a mean distance value, calculaded over the entire extent? Is it an average value of the distance calculated in correspondence of reference objects or areas? Moreover, how the presence/absence of GPS-measured Ground Control points can affect the final accuracy and usability of the results?

Corrections needed in the text:

Page 11, Line 2: replace "determining" with "determine". Page 16, Caption figure 10: replace "doted" with "dotted". Page 18, line 22: add "using" between "obtained" and "the".

---

## Referee Comment (RC2) · Anonymous Referee #2 · 16 Mar 2018

The paper "Usability of aerial video footage for 3D-scene reconstruction and structural damage assessment" provides a very good insight on the use of video data to assess structure damages in 2D and 3D. Two datasets are used here, corresponding to earthquakes of Tainan 2016 and Pescara del Tronto 2016. Cutting-edge techniques are explored for the purpose of pots-earthquake structural damage assessment. The complete processing is complex, and many options are considered: oblique vs nadir, resolutions, phots vs videos, random vs wise frame selection, etc. It's of course impossible to test all the combinations and one can also expect is that each input dataset has different properties. The paper is already dense in information, and by the way very well written. It gives some promising views on the potential of video footage for natural hazards and disaster analysis. I have then only some minor corrections to propose:

[Figure]

- Commercial softwares have been used (Pix4D, 3DFlow). People working with SfM know that various processing packages produces various results. Can you comment on that in the discussion – how do you think that results would have been different if other packages would have been used?

- WFS, RFS and IQI refers to 3DFlow functionmalities – Please explain shortly the principles and provides references.

- Please provide some extra information on the CNN model used (to avoid to have read Vetrivel et al 2017)

- How long would be the full process in an operational context? from the raw video footage arrival to a SDA

---

## Author Comment (AC1) · 25 Mar 2018

Usability of aerial video footage for 3D-scene reconstruction and structural damage assessment - Author response

Dear Referee,
Firstly, thank you for your valuable time and contribution. The comments given will definitely help improving the quality of this research paper. The responses to your comments are hereunder detailed point-by-point, and are complemented by a marked-up and a corrected manuscript version.

[Figure]

1. Probably reducing the frame redundancy using the WFS for video data can generate a lack of important photos, needed for an accurate Point Cloud calculation. For that reason, the video results with WFS are generally worse than the RFS. Maybe the WFS method is not appropriate in this case because the photos selected as good were in a non-optimal position. Is clear that the results of a Point Cloud generation are strongly related to the overlap and sidelap percentage between photograms. Maybe with different rules for WFS the results could be even better than using the RFS, even if the latter method is certainly faster.

   **Response.** There are different video frame selection approaches which are based on weights and thresholds (Hasegawa et al., 2000; Ahmed et al., 2010; Alsadik et al., 2013); however, it is very complex to define an optimal strategy that works in all conditions. WFS method is an approximation, which ensures having a sufficient number of frames for the 3D reconstruction, selected manually based on their IQI and redundancy. Different results are expected using more elaborated methods, but WFS method can work quite well in most of the practical cases.

   **Modifications.** This is now explained in **page 20 line 5**: "The use of a more elaborated approach might result in more accurate 3DPCs; however, the variability of data and external characteristics limits the development of an optimal frame selection approach that can deal with all conditions".

2. Considering 3D Internal Accuracy assessment. The authors use planar fitting as a method to assess the accuracy of the resulting 3DPCs. It seems that only one planar object was used to perform this analysis. Maybe, considering more planar objects preferably equally distributed, for example, with respect to shaded and well-lighted areas or central or peripheral position could lead to more robust conclusions.

   **Response.** More objects were analyzed during the research; however, results were statistically similar and due to the paper extent only the most meaningful

experiments are shown.

3. Considering 3D External Accuracy: it is not clear how the distance between each 3DPC and the reference one is considered. Is it a mean distance value, calculated over the entire extent? Is it an average value of the distance calculated in correspondence of reference objects or areas? Moreover, how the presence/absence of GPS-measured Ground Control points can affect the final accuracy and usability of the results?

   **Response.** It refers to the mean distance to the reference 3DPC, calculated from all 3D points distances to the reference 3D points. Besides, the presence/absence of Ground Control Points (GCPs) is a relevant parameter, because video data in most cases do not hold accurate geo-positional information and depend either on GPCs or an accurate 3D model. Moreover, the geo-localization of video frames is limited by their oblique perspective and resolution. Low quality GCPs can affect the geometrical accuracy of the video-generated 3DPCs and also the depictability of damage-related geometrical features.

   **Modifications.** 3DPC external accuracy is now better explained in **page 9 line 31**: "the mean distance of the 3D points to the reference 3D points was computed to determine every 3DPC external accuracy", and also in **page 13 line 6**: "Independent oblique and nadir photos-based 3DPCs registered high accuracies, with mean distances of 2 and 4 cm to the reference model, respectively" and also **Page 15 Figure 6, Y axis**: "Mean distance to the reference model".

We hope all the comments were sufficiently clarified and corrected. We remain at your disposal for any additional comment or modification you might deem necessary.

Kind regards,
The authors

Please also note the supplement to this comment:

https://www.nat-hazards-earth-syst-sci-discuss.net/nhess-2017-409/nhess-2017-409-AC1-supplement.pdf

---

## Author Comment (AC2) · 25 Mar 2018

Usability of aerial video footage for 3D-scene reconstruction and structural damage assessment - Author response

Dear Referee,

Firstly, thank you for your valuable time and contribution. The comments given will definitely help improving the quality of this research paper. The responses to your comments are hereunder detailed point-by-point, and are complemented by a

marked-up and a corrected manuscript version.

1. Commercial software have been used (Pix4D, 3DFlow). People working with SfM know that various processing packages produces various results. Can you comment on that in the discussion – how do you think that results would have been different if other packages would have been used?
   **Response.** We used two well known packages that are commonly used by many research groups. These software provide state-of-the-art results. The use of conventional instruments was decided on purpose in order to be closer to the "normal" processing of data. Similar results could be generated with other software resources like AgiSoft and Photoscan Pro, or open source packages such as ColMap, MicMac, etc. Papers like Remondino et al. (2012) and Gross (2015) have already discussed the different performances of these software, and differences were not significant.
   **Modifications.** The explained is now added in **page 18 line 34**: "Similar results are expected with the use of other commercial software (e.g. Agisoft, Photoscan, etc.) or open source packages (e.g. ColMap, MicMac, etc.)".

2. WFS, RFS and IQI refers to 3DFlow functionalities – Please explain shortly the principles and provides references.
   **Response.** IQI functionality is explained in **page 8 line 8**: "Image Quality Index (IQI) indicate possible sources of error for image-based 3D scene reconstruction, such as the presence of low-texture areas or motion-blur effects (3D Flow, 2017)". RFS does not refer to 3DFlow functionalities; however, it is also described in **page 5 line 13** as a method "only defined by an empirical number of randomly-selected frames (defined in Pix4D)".
   **Modifications.** Wise Frame Selection (WFS) functionalities are now shortly described in **page 6 line 2**: "WFS uses as guideline each frame initial 3D position (generated with Pix4D (2017)) and their correspondent Image Quality Index (IQI)

value (computed with 3D Zephyr 3DFlow (2017)), with the aim of discarding the most redundant (i.e. more than 80% overlap) and lowest quality (i.e. an IQI lower than 0.5) frames, respectively.".

3. Please provide some extra information on the CNN model used (to avoid to have read Vetrivel et al 2017)
**Modifications.** In **page 9 line 2** more information about the CNN model is now provided: "A deep learning approach presented by Vetrivel et al. (2017) was tested. This is in based on the 'imagenet-caffe-alex' (Krizhevsky et al., 2012) CNN model which is composed of different groups of layers. Convolutional layers represent the first group, and are a set of filter banks composed of image and contextual feature filters. The following group corresponds to data shrinking and normalization layers. Finally, the last group transforms all the information generated and outputs features with high-level reasoning; usually this layer is connected to a loss function for the final classification. This approach uses a large amount of training samples (i.e. labelled images) to tune the weights of the CNN classification layer."

4. How long would be the full process in an operational context? from the raw video footage arrival to a SDA.
**Response.** The present research aims at comparing video data and aerial photos for SDA, and not implicitly at implementing a real-time procedure. Nevertheless, from the analysis performed some limitations are evident, mainly for a rapid video-based 3D reconstruction and damage-feature depictability. These are for example: The lack of proper 3D GCPs for and efficient geo-registration, the lack of pre-existing data, a software-integrated operational flow and an approach to aggregate all the 3D damage-related features.

We hope all the comments were sufficiently clarified and corrected. We remain at your disposal for any additional comment or modification you might deem necessary.
Kind regards,
The authors

Please also note the supplement to this comment:
https://www.nat-hazards-earth-syst-sci-discuss.net/nhess-2017-409/nhess-2017-409-AC2-supplement.pdf

———————————————————